# Biomedical Applications of Sulfonylcalix[4]arene-Based Metal–Organic Supercontainers

**DOI:** 10.3390/molecules29061220

**Published:** 2024-03-08

**Authors:** Ya-Wen Fan, Meng-Xue Shi, Zhenqiang Wang, Feng-Rong Dai, Zhong-Ning Chen

**Affiliations:** 1College of Chemistry and Materials Science, Fujian Normal University, Fuzhou 350007, China; fanyawen@fjirsm.ac.cn (Y.-W.F.); shimengxue@fjirsm.ac.cn (M.-X.S.); 2State Key Laboratory of Structural Chemistry, Fujian Institute of Research on the Structure of Matter, Chinese Academy of Sciences, Fuzhou 350002, China; 3Department of Chemistry, Center for Fluorinated Functional Materials, University of South Dakota, 414 East Clark Street, Churchill-Haines Laboratories, Room 115, Vermillion, SD 57069-2390, USA; zhenqiang.wang@usd.edu; 4University of Chinese Academy of Sciences, Beijing 100049, China

**Keywords:** sulfonylcalix[4]arene, coordination cages, drug delivery, biomedical applications, host–guest chemistry, metal–organic supercontainers

## Abstract

Coordination cages sustained by metal–ligand interactions feature polyhedral architectures and well-defined hollow structures, which have attracted significant attention in recent years due to a variety of structure-guided promising applications. Sulfonylcalix[4]arenes-based coordination cages, termed metal–organic supercontainers (MOSCs), that possess unique multi-pore architectures containing an *endo* cavity and multiple *exo* cavities, are emerging as a new family of coordination cages. The well-defined built-in multiple binding domains of MOSCs allow the efficient encapsulation of guest molecules, especially for drug delivery. Here, we critically discuss the design strategy, and, most importantly, the recent advances in research surrounding cavity-specified host–guest chemistry and biomedical applications of MOSCs.

## 1. Introduction

Discrete coordination cages are a fascinating class of synthetic supramolecular receptors, which possess well-defined internal cavity structures for accommodating guest molecules [1,2,3,4,5,6]. Their built-in internal cavities can bind suitable guest molecules; thus, they provide specific chemical microenvironments to promote a large range of applications in gas storage and separation [7,8,9], molecular recognition [10,11,12], stabilization of reactive species [13,14,15], supramolecular catalysis [16,17,18], biomedical applications [19,20,21], etc. Thanks to the highly directional and relatively rigid nature of metal–ligand bonds, they are accessible through the coordination-driven self-assembly of suitable metal ions and multidentate organic ligands [22,23]. They usually adopt well-organized geometrical shapes and were also known as metal–organic polyhedrons (MOPs), metal–organic cages (MOCs), metallacages, or porous coordination cages (PCCs).

Although significant progress has been made in the design and application of coordination cages, the characteristics of containing only single binding domains and the unsatisfactory biostability and biocompatibility of traditional coordination are among the major drawbacks that have hampered their wider application in the field of biomedicine. In the past decade, an emerging class of coordination cages, termed as metal–organic supercontainers (MOSCs), have been attracting increasing attention in view of their robust architectures, predictable synthetic procedures, and unique structural features [24,25]. The MOSCs are conveniently accessible from the self-assembly reactions of sulfonylcalix[4]arene container precursors [26,27,28], with metal ions and suitable carboxylate linkers. They have been demonstrated to exhibit multi-pore architectures containing a well-defined *endo* cavity and multiple *exo* cavities that imitate the structural topology of virus capsids; hence, they provide multiple binding domains for efficient loading of multiple drug molecules. Furthermore, their distinguishing features (excellent thermal and chemical stability and suitable molecular size (<10 nm)) make discrete MOSCs excellent candidates for drug delivery carriers in biomedical applications.

Numerous reviews have provided a comprehensive overview of the design and construction of coordination cages for biological applications [20,29,30,31,32]. At present, some critical reviews on the synthesis and application of MOSCs have been published [24,25,33]; however, there is no review exclusively focusing on the bio-applications of MOSCs. Thus, this article gives a brief introduction to MOSCs and their host–guest chemistry and highlights significant recent developments in biomedical applications of MOSCs. We aim to provide a clear and comprehensive understanding for this exciting research field to promote further development and applications.

## 2. Structural Regulation and Cavity Engineering of MOSCs

The molecular topologies of MOSCs are easily and controllably modulated by geometries of carboxylate linkers (Figure 1). The assembly of rigid tricarboxylate ligands, such as 1,3,5-benzenetricarboxylic acid (H_3_BTC) [34,35], with metal ions (i.e., Co^2+^, Ni^2+^, or Mg^2+^) and *p-tert*-butylsulfonylcalix[4]arene (H_4_TBSC), gave rise to the face-directed octahedral MOSCs (namely Type-I MOSCs, Figure 1) [36,37,38,39,40]. These consist of six sulfonylcalix[4]arene-supported tetranuclear clusters residing on the vertices and eight tricarboxylate linkers on the triangular facets of the octahedron. This type of MOSC generate seven well-defined cavities, including one large *endo* cavity surrounded by tricarboxylate ligands and six *exo* cavities based on the upper rim of sulfonylcalix[4]arenes; these can provide multiple binding domains that are suitable for efficient guest encapsulation. Moreover, the size of the *endo* cavity of MOSCs is tunable by simply increasing the length of the tricarboxylate connectors. The single-arm elongated tricarboxylate ligand, 5-[(4-carboxybenzyl)amino]-isophthalic acid, obtained from partially extending the regular BTC ligand with a benzylamino spacer, provided a tetragonal–elongated octahedral MOSC; this had an *endo* cavity with a volume that increased from 0.55 nm^3^ to 1.00 nm^3^ [41]. Furthermore, the volume of the *endo* cavity was significantly enlarged from 0.55 nm^3^ to 2.75 nm^3^ by simply changing the tricarboxylate ligand from BTC to 4,4′,4″-benzene-1,3,5-triyl-tribenzoic acid (BTB) [34,36,37].

Replacing the tricarboxylate linkers with a linear dicarboxylate ligand of 1,4-benzenedicarboxylate (BDC), the edge-directed octahedral MOSCs (Type-II, Figure 1) are readily constructed [42]; here, six [M_4_(TBSC)(μ_4_-H_2_O)] tetranuclear units are located on the vertices of the octahedron, while twelve dicarboxylate ligands ride on the edges of the octahedron. As a result, the edge-directed octahedral MOSCs have one *endo* cavity and six *exo* cavities, similar to the face-directed octahedral MOSCs; however, they provide remarkably wider apertures, which are located on the faces of octahedrons. In comparison with BTC-based MOSCs, 1,4-benzenedicarboxylate (BDC)-based MOSCs possess a significantly larger *endo* cavity volume (1.2 nm^3^ vs. 0.55 nm^3^), even though both MOSCs have a comparable outer diameter (*ca.* 3 nm). Furthermore, the sizes and functionalities of the *endo* cavities of Type-II MOSCs can be fine-tuned by modifying the dicarboxylate ligands [42,43,44].

When rigid angular dicarboxylate ligands were employed as the linkers, the self-assembled reactions with suitable metal ions and sulfonylcalix[4]arenes resulted in the formation of a new type of MOSC (Type-III, Figure 1) with a barrel-shaped topology [45,46,47]. 1,3-benzenedicarboxylate (1,3-BDC)-directed Type-III MOSCs are composed of four tetranuclear units that are double-bridged with eight 1,3-BDC linkers; these possess four *exo* cavities and a barrel-shaped *endo* cavity, featuring two wide openings (*ca.* 7 Å × 7 Å, defined by the shortest distance between two opposite carboxylate linkers), a wider inner dimension (*ca.* 10 Å × 10 Å, defined by the longest distance between two opposite carboxylate linkers), and a deep pocket (with a depth of *ca.* 12 Å) [45]. Additionally, the host–guest chemistry of Type-III MOSCs can be achieved through functionalization on the 5-position of the 1,3-benzenedicarboxylate ligand. A 5-sulfo modification on 1,3-BDC provided an anionic MOSC {[Ni_4_(TBSC)(μ_4_-H_2_O)]_4_(5-SO_3_-1,3-BDC)_8_}^8−^ that showed higher binding capacity toward MeOH gas in the solid state and a stronger binding affinity toward methylene blue (MB) in solution in comparison with its 1,3-BDC-based neutral counterpart [45].

A new family of MOSCs (Type-IV, Figure 1), using angular flexible dibenzoate ligands containing methylene, methyleneamino, or bismethyleneamino as spacers, have been developed by Wang and coworkers [48]. Type-IV MOSCs contain two tetranuclear units that join together with four V-shaped dicarboxylate linkers to form a cylindrical molecular topology. The incorporation of various functional spacers in the flexible dicarboxylate linkers brings about a diverse range of supramolecular functions, including selective guest binding for molecular recognition and separation; these have potential catalytic applications [48,49].

In addition to the regular Type I–IV MOSCs, there are some other less common MOSCs [50,51,52]. Sheng and coworkers reported a trigonal antiprismatic MOSC (Figure 1) [41] using single-arm-lengthened tricarboxylate ligand as a linker, together with cyclohexane-1,3,5-tricarboxylate (CTC) ligand as secondary linker; this MOSC had a cylinder-shaped *endo* cavity with a radius and a height of *ca.* 9 and 15 Å and volume of *ca.* 1 nm^3^. A trigonal prismatic MOSC [53] using a 2,5-thiophenedicarboxylate (TDC) ligand as linker was isolated by Liao and coworkers; this MOSC contains an *endo* cavity with dimensions of about 12.1 × 12.1 × 9.6 Å^3^.

In addition to the modulation of the *endo* cavities of MOSCs using carboxylate linkers, the *exo* cavities of MOSCs are also tunable via chemical functionalization on the *para* substituent groups of the sulfonylcalix[4]arenes [42]. Cheng and coworkers reported the modulation of the *exo* cavities of Type-IV MOSCs by the *para*-modification of sulfonylcalix[4]arene with phenyl units; this resulted in MOSCs with much wider and deeper *exo* cavities and additional π–π interactions between the adjacent cage molecules that enhance the robustness of crystal packing; these additional interactions promoted the MOSC’s overall porosity, gas/vapor adsorption capacity, and selectivity in the solid state [54].

## 3. Biomedical Applications of MOSCs

### 3.1. Host–Guest Chemistry and Drug Encapsulation

MOSCs feature a dual-cavity structure containing both *exo* and *endo* cavities that provide multiple binding domains. The cavities are capable of encapsulating specific guest molecules, such as drug molecules, through non-covalent interactions, including electrostatic interactions, π–π interactions, hydrophobic interactions, and hydrogen bonding. Dai and coworkers [55] investigated the cooperative binding and stepwise encapsulation of drug molecules using a BTC-based MOSC ({Co_4_(TBSC)(H_2_O)}_6_(BTC)_8_, namely, **MOSC-1-Co**) (Table 1) [34] as the host and (*R*)-(+)-rabeprazole sodium (**D1**) and (*S*)-(−)-pantoprazole sodium (**D2**) as the guests (Figure 2a). **MOSC-1-Co** features a well-defined *endo* cavity with an inner diameter of *ca.* 1.4 nm and six open-ended *exo* cavities; these are accompanied by eight external pockets surrounded by three adjacent TBSC units and BTC linkers. UV–Vis spectroscopic titration experiments carried out in CHCl_3_ solutions were used to confirm the interesting cooperative binding and stepwise encapsulation of drug molecules by **MOSC-1-Co**. Upon the gradual addition of drug solutions to an **MOSC-1-Co** solution in CHCl_3_, clear changes in the absorption spectra of **MOSC-1-Co** were observed; here, the intensity of the maximum absorption at 350 nm was seen to increase gradually, along with the appearance of a shoulder centered at 370 nm. These observations indicated the successful encapsulation of drug molecules within the cavities of **MOSC-1-Co**. As shown in Figure 2, **MOSC-1-Co** quickly traps two equivalents of the drug molecules inside the *endo* cavity with strong binding constants of 3.04 × 10^5^ and 2.81 × 10^5^ M^−1^ for **D1** and **D2**, respectively. Each of the six bowl-shaped *exo* cavities subsequently catch a drug molecule with medium binding affinities of 5.15 × 10^4^–9.60 × 10^4^ M^−1^. Further encapsulating of drug molecules can be achieved by the wide-opening external pockets with weaker binding strengths of 0.84 × 10^4^ M^−1^ and 1.69 × 10^4^ M^−1^ for **D1** and **D2**, respectively. The in-depth understanding of the cooperative host–guest chemistry of MOSCs affords new avenues for the design and development of synthetic receptors for truly biomimetic functional applications (Table 1).

Liao and coworkers [36,37] utilized tripodal 1,3,5-benzene-tricarboxylate (BTC), 4,4′,4″-benzene-1,3,5-trial-tribenzoate (BTB), 4,4′,4″-benzene-1,3,5-trityl-tris-(ethyne-2,1-diyl)-tribenzoate (BTE), and 4,4′,4″-(benzene-1,3,5-trityl)-tris(benzene-4,1-dial)tri-benzoate (BBB) as linkers to form a series of MOSCs, namely CIAC-105, CIAC-106, CIAC-107, and CIAC-114, respectively. These MOSCs adopt face-directed octahedral architectures, and the volumes of their *endo* cavities are significantly increased from CIAC-105 to CIAC-114, with the increasing molecular size of the tricarboxylate linker from BTC to BBB. CIAC-114 features a peripheral diameter of *ca.* 5.4 nm, an inner diameter of up to 2.7 nm, and twelve fusiform apertures measuring *ca.* 2.0 nm × 1.1 nm. After activation by supercritical CO_2_, the drug loading and releasing experiments of CIAC-106 and CIAC-114 were evaluated using an anti-inflammatory drug, Ibuprofen (Ibu), as the guest; these were monitored by IR, NMR, and high-performance liquid chromatography (HPLC). Since the Ibu molecules have good solubility in hexane, while the activated containers are insoluble in hexane, the solid–liquid adsorption experiments were set up by screening various conditions of mass ratio (*w*/*w*) between Ibu and MOSCs, the contact time, and the initial concentration of Ibu. The successful encapsulation of Ibu into the cavities of MOSCs was confirmed by FT-IR, 2D-diffusion-ordered NMR spectroscopy (DOSY), the measurements of BET surface areas, and the TG studies. The Ibu loading capacities were evaluated to be 0.44 g Ibu per gram for desolvated CIAC-106 sample and 0.31 g Ibu per gram for desolvated CIAC-114. Finally, the drug release behaviors of Ibu-loaded MOSCs were conducted by dialyzing the Ibu-loaded MOSC samples against a phosphate-buffered saline (PBS) solution (pH = 7.4), as monitored by HPLC. For the control of the TBSC system, 67% of the loaded drug was released within 7 h, whereas around 26% and 34% of the loaded drug was released within 45 h for CIAC-106 and CIAC-114, respectively. The results demonstrated that MOSCs with a high drug-loading capacity and a tunable drug-release process could be desirable candidates for long-term drug-delivery systems.

A multiple-guest co-loaded MOSCs-based drug-delivery system (MgDHIA) has been reported by Xu, Dai, Zhang, and coworkers [56]; they utilized the 4,6-dihydroxyisophthalate ligand (DHIA) as the angular linker. MgDHIA adopts the typical geometry of a Type-III MOSC, possessing an outer dimension of *ca.* 2.9 nm × 2.9 nm × 1.3 nm, a wide, barrel-shaped *endo* cavity of *ca.* 1.0 nm × 1.0 nm with two wide openings, and four cup-shaped *exo* cavities with diameters of *ca.* 0.7 nm. Introducing hydroxyl groups on the 4,6-positions of 1,3-BDC can improve the aqueous solubility and stability of MgDHIA, while providing additional hydrogen bonding donors to enhance the binding affinity toward the drug molecules. The host–guest chemistry studies have been performed well in the homogeneous solution by the NMR spectroscopic titration technique. The systemic ^1^H NMR and DOSY studies clearly demonstrated that MgDHIA is able to efficiently encapsulate homo-guest molecules through its *endo* and *exo* cavities. It is also capable of selectively capturing the anti-inflammatory drug rapamycin (RAPA) within the *endo* cavity, and subsequently encapsulating the targeting agent (folic acid, FA) and/or fluorescent labeling agent (Rhodamine B, RB) into the *exo* cavities. Thus, these MOSCs form unique multiple-drug co-loaded systems named RAPA-FA@MgDHIA or RAPA-FA-RB@MgDHIA (Figure 3a,b). The stability and drug-release kinetics of the RAPA-FA@MgDHIA system in the PBS buffer at pH values of 7.4, 6.0, and 5.0 were further examined by UV–Vis spectroscopy, monitored at different time intervals. The different pH values of 7.4, 6.0, and 5.0 mimic the different environments of cytosol, lysosome, and inflammation, respectively. The release profiles demonstrated the fast stepwise release processes of RAPA molecules from *exo* cavities, occurring within *ca.* 10 h and then within the *endo* cavity within 5 days; meanwhile, the release of FA molecules occurred immediately and was almost complete after 12 h. The increase in the acidity of the PBS buffer helps in speeding up the release process of drug molecules; this confirmed the idea of the pH-responsive drug-releasing behavior in an inflammatory environment. MOSCs must have good biocompatibility and low biotoxicity when used as a drug-carrying system for clinical treatment. In the cellular test results of MgDHIA, the cells still showed good bioactivity, even when the concentration of the coordination cage as high as 20 μg/mL (Figure 3c). This suggests that MgDHIA has excellent biocompatibility and low biotoxicity, making it a good potential option for clinical therapeutic applications.

Dai and coworkers [57] reported the design and assembly of a new MOSC (ZnPMTC) from a flexible tetracarboxylate linker, 5,5′-(*p*-phenylene-bis(methanamino)-di-isophthalic acid (H_4_PMTC); here, two isophthalic acid units were connected by a *p*-phenylene-bis(methanamino) (−NHCH_2_−Ph−CH_2_NH−) spacer. ZnPMTC is composed of four Zn_4_TBSC tetranuclear moieties and four PMTC linkers, adopting a barrel-shaped molecular structure with one cylinder-shaped *endo* cavity and four bowl-shaped *exo* cavities (Figure 4a). The outer dimensions are calculated to be *ca.* 2.9 nm × 2.9 nm × 1.8 nm. The introduction of the *p*-phenylene-bis(methanamino) group into the PMTC ligand provides additional hydrogen bond donors, which provide the benefit of enhancing the guest binding affinity; also, this produces two adjustable apertures, resulting from the free rotation of the phenylene groups. This is applicable for pH-responsive guest-release process. The NMR titration experiments using three anti-inflammatory drugs (naproxen, NPX; diclofenac sodium, DCF; aspirin, AP) showed that the guests demonstrated stepwise drug-loading behaviors in ZnPMTC through sequential binding at the *endo* and *exo* cavities. The drug-release processes were subsequently evaluated in PBS solutions at pH values of 7.4, 6.0, and 5.0, monitored by UV–Vis spectroscopy. The results suggested that the drug-loaded system can rapidly release drugs from the *exo* cavities but there is a slow release from the *endo* cavity, accompanying the remarkable pH-dependent effect (Figure 4b). To verify the clinical practicality of the drug vector, the biocompatibility, the cellular uptake, and the anti-inflammatory effect on the macrophages of the drug-loaded systems have been further studied. At a safe dose of 10 μg/mL, the drug-loaded ZnPMTCs enter the cells by cellular internalization and release the drug molecules through the endolysosomal pathway (Figure 4c).

### 3.2. Ion/Molecule Recognition

In addition to drug loading for therapeutic effects, assays of a diagnostic nature are equally essential components of medical treatment. Of particular note is the ability of certain MOSCs to produce stimulatory responses to specific pathogenic substances. Introducing both a proton binding site (-NH-) and a fluorescent tag (pyrenyl) into the 5-position of 1,3-BDC ligand, two Type-III MOSCs, namely Co-NH-pyr and Zn-NH-pyr [46], were developed (Figure 5a). Both MOSCs feature barrel-shaped *endo* cavities and shuttlecock-like tetranuclear M_4_(TBSC) *exo* cavities. They exhibit obstructed window openings with pyrenyl groups assembled via the edge to form π–π stacking. These MOSCs exhibit an exceptional proton binding capacity (over 50 equiv.) and an interesting proton-dependent photoluminescence switching behavior, which is attributed to the presence of both amino and pyrenyl moieties of the supercontainer constructs (Figure 5b). The multiple proton binding pathways are accessible, and the extraordinary binding capability of the MOSCs emphasizes their great potential to act not only as proton receptors but also as potential proton modulators via the organized release or removal of the protons.

Polycyclic aromatic hydrocarbons (PAHs) are carcinogenic and may cause immunosuppression when they enter the human body. The detection of urinary 1-hydroxypyrene (1-OHP) is a useful and reliable marker for assessing human exposure to PAHs. To overcome the shortages of traditional testing methods, Dai and coworkers [59] creatively designed a binuclear terbium-sulfonylcalixarene-based coordination cage for the conventionally selective detection of urinary 1-hydroxypyrene. The dinuclear terbium(III) complex, **Tb-TBSC**, was obtained from the reaction of Tb^3+^ and H_4_TBSC; here, a characteristic Tb^III^-centred luminescence is generated through a ligand-to-metal energy transfer process from the efficient antenna ligand TBSC to the Tb^3+^ ions. Moreover, the TBSC units also provide a suitable cup-shaped cavity to encapsulate the 1-OHP to form a stable host–guest complex that exhibits a remarkable fluorescence-quenching phenomenon due to the enhancement of the host–guest intermolecular charge transfer; in turn, this reduces the ligand-to-metal energy transfer process. The **Tb-TBSC** luminescent sensor shows quick, sensitive, and selective response to 1-OHP; thus, it provides a new sensing platform for the clinical diagnosis of human exposure to PAHs (Figure 6).

### 3.3. Targeted Drug Delivery and Therapeutic Effects

Compared with traditional drug-delivery methods, the use of MOSCs to deliver drugs has significant advantages, as it enables the targeted release of drugs, thus effectively reducing drug abuse and minimizing side effects. Zhou and coworkers designed and synthesized two functionalized Type-I MOSCs, PCC-1 and PCC-2, to investigate the subcellular-compartment-targeting effect of the porous coordination cages [38,60]. PCC-1 is a neutral octahedral cage that consists of six tetranuclear Zn_4_TBSC clusters as the vertexes and eight panel ligands as the faces. PCC-1 possesses an intrinsic fluorescence property inherited from the panel ligand; this allows for convenient monitoring of its cellular transportation process. PCC-2 adopts a similar octahedral topology to that of PCC-1 but gives an anionic host with a total of 30 negative charges due to the sulfate functionalization on the *para*-position of TBSC. Using an anti-tumor drug ((+)-Camptothecin, CPT) or cationic fluorescent dye (Rhodamine B, RB) as the cargo, the cargo encapsulation and release behaviors of PCC-1 and PCC-2 were further investigated. The loading capacities were then evaluated to be 0.110 g/g for CPT@PCC-1 and 0.076 g/g for RB@PCC-2 through *endo* encapsulation. The cargo-release experiments revealed that CPT@PCC-1 and RB@PCC-2 may remain stable for several hours under neutral conditions and the capsule disassembly and subsequent drug release may then occur at a constant rate (Figure 7a). The in vitro experiments revealed that CPT@PCC-1 had lower cytotoxicity for normal cells, and mainly accumulated in the nucleus after entering the cells; this is in contrast to free-drug CPT that is mainly distributed in the cytoplasm. The highly negative complex RB@PCC-2 was distributed evenly in the cytoplasm, while the free RB exhibited clear mitochondrial binding. Anticancer efficacy studies revealed the effective nucleic delivery of CPT@CPP-1, giving rise to a significantly lower viability of 9.5% for HeLa cells; meanwhile, the CPT@PCC-2 and CPT@PCC-3 groups remained above 60% cell viability (Figure 7b). From the confocal laser scanning microscopy images presented in Figure 5b, we can observe the release process of PCC-1- and PCC-2-encapsulated drug molecules. Unlike free CPT drug molecules, which are mainly distributed in the cytoplasm, PCC-1 can encapsulate CPT drug molecules and transport them to the nucleus of the target. This enables CPT drugs to exert their drug activity in the nucleus and inhibit gene expression in cancer cells, thereby enabling cancer therapy. In contrast, RB encapsulated by PCC-2 was mainly distributed in the cytoplasm. This suggests that the electrically neutral ligand cage PCC-1, constructed from specific ligands, can encapsulate specific drug molecules and achieve the targeted release of drug molecules.

MOSCs can target the nucleus and accurately accumulate in specific cellular environments, such as the acidic environment in which inflammatory cells reside. Taking advantage of the excellent biocompatibility and low toxicity of MgDHIA, the multiple-drug co-loaded system of RAPA-FA@MgDHIA [56] was further explored for the treatment of temporomandibular joint (TMJ) inflammation. The in vitro tests revealed that MgDHIA and RAPA-FA@MgDHIA have satisfactory biocompatibility with a safe dose up to 10 μg/mL, excellent cellular uptake behaviors, an outstanding macrophage-targeting effect, and stepwise drug release via the acidic endo–lysosomal pathway. The in vitro therapeutic experiments performed on polarized macrophages in an inflammatory environment demonstrated that the RAPA-FA@MgDHIA repolarized M1 macrophages to M2 macrophages by inhibiting the activation of the NF-kB pathway. The in vivo studies suggested that RAPA-FA@MgDHIA efficiently accumulated in inflamed synovial tissues to alleviate synovial inflammation; additionally, it protected the articular cartilage and subchondral bone structures. Moreover, no histologic damage was observed after the H&E staining of major organs and the excretion of RB-FA@MgDHIA was achieved through the intestines (Figure 8).

MOSCs also enable the targeting of mitochondria in inflammatory cells. Recently, Xu, Dai, Zhang, and coworkers established an efficient supramolecular nanocarrier for endo/lysosomal escape and mitochondrial targeting; they utilized the Zn-based MOSC, Zn-NH-pyr [46], as a proton sponge, electron reservoir, and reactive oxygen species (ROS) scavenging agent [58]. Zn-NH-pyr displays excellent extra- and intracellular ROS scavenging activity (Figure 9), which is attributed to its inherent and specific structural characteristics. These include the catalytic centers of Brønsted acid μ_4_-H_2_O sites and –NH– active sites, the well-defined *endo* cavity to capture and enrich radicals, and the extraordinary electron-donating and -accommodating capability. Zn-NH-pyr, with a significantly high proton binding capacity (up to 50 equiv.), enables itself to escape from the endo/lysosome and target the mitochondria. Zn-NH-pyr showed excellent performance in terms of biotoxicity, which was extremely low. From the live/dead cell staining graphs, it can be observed that the cells remained active even when the concentration of Zn-NH-pyr reached 20 μg/mL. This property indicates that this ligand cage has excellent potential and broad application prospects in clinical practice. After loading the drug 4-octyl itaconate (4-OI) inside the *endo* and *exo* cavities, 4-OI@Zn-NH-pyr can serve as a synergistic therapeutic system for inflammatory macrophages and osteoclasts and be safely excreted from the body through the kidney thanks to the unique properties of good solubility and ultrasmall molecular size (<3 nm) (Figure 9).

## 4. Conclusions and Outlook

As a subset of synthetic receptors, sulfonylcalix[4]arene-based coordination cages, also known as discrete metal–organic supercontainers (MOSCs), comprise a growing and diverse area of study in supramolecular disciplines. In this review, recent important developments surrounding MOSCs in biomedical applications are presented. The advantages of MOSCs in biomedical applications include the following: (i) MOSCs are synthesized in a straightforward and convenient way from coordination-driven self-assembled reactions of shuttlecock-like sulfonylcalix[4]arene units, metal ions, and carboxylate linkers. Such a modular synthetic strategy is beneficial for the structural and functional regulation of MOSCs through the modification of each component. (ii) MOSCs possess distinct multiple binding domains of *endo* and *exo* cavities that are capable of the cooperative loading of multiple drugs. On the one hand, the size and shape of the *endo* cavity are tunable by manipulating the length and configuration of the carboxylate ligands. On the other hand, the *exo* cavity can be easily modulated by chemically modifying the *para* substituents of the sulfonylcalix[4]arenes. (iii) The targeted transportation of drug-loadable MOSCs across different subcellular barriers can be achieved through the modulation of the electronic property of MOSCs by incorporating anionic groups or the proton sponge effect. (iv) MOSCs have good thermal and chemical stability, ensuring that MOSCs deliver and accumulate therapeutic drugs in targeted sites and can be safely cleared from the system through excretion. (v) The intrinsic catalytic activity of MOSCs provides a possibility for their application in regulating the levels of intracellular free radicals, such as reactive oxygen species.

As an emerging technology in the field of biomedical applications, MOSCs-based drug-delivery nano-systems have a bright and exciting future. Recent advancements have been focusing on utilizing MOSCs as drug carriers in medical treatment. The controllable release of drugs in MOSC-based drug-delivery systems in response to stimuli, such as temperature, enzymes, light, or ultrasound, still requires significant improvements. In addition, although MOSCs enable the construction of targeted therapies and multiple drug-delivery systems, they still require significant validation before clinical applications can commence. With a deeper understanding of diseases, future medical treatment will pay more attention to individualized treatments, and this drug-carrying nano-system is expected to realize personalized drug delivery, improve therapeutic effects, and reduce side effects by precisely targeting specific cells or tissues. In summary, developing MOSCs-based drug-carrying nano-systems needs to address technical and scientific issues and strengthen interdisciplinary cooperation to ensure their safety and efficacy to serve the future of healthcare better.

## Figures and Tables

**Figure 1 molecules-29-01220-f001:**
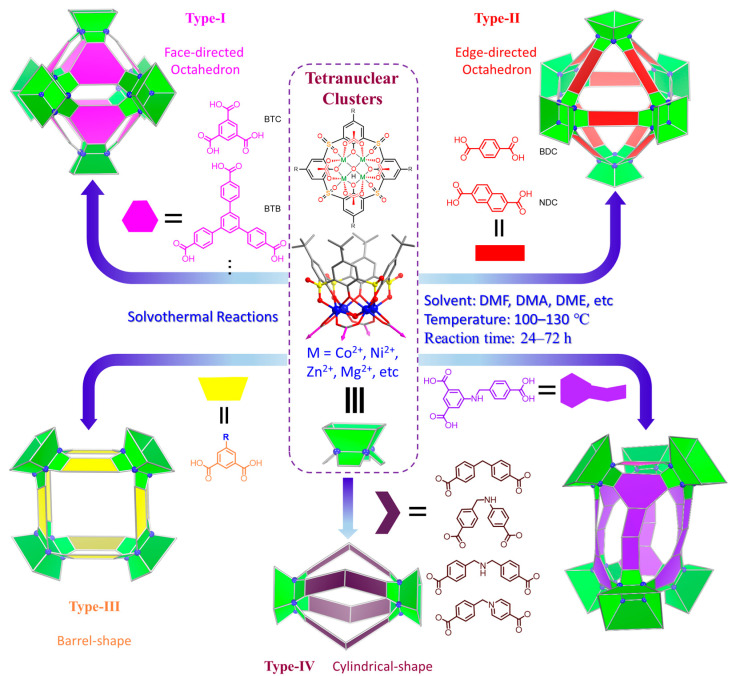
Structural regulation of MOSCs by judicious design of various carboxylate linkers.

**Figure 2 molecules-29-01220-f002:**
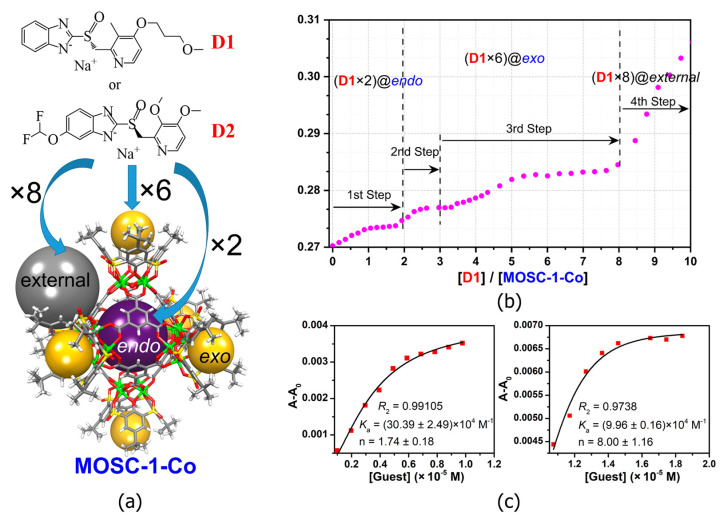
(**a**) The structures of **MOSC-1-Co** and drug guests (**D1** and **D2**), and the processes of drug encapsulation. The Co^2+^ centers are shown in green. The spheres serve to guide the eyes representing the *endo* cavity (purple), six *exo* cavities (yellow), and one of the eight external pockets (gray). (**b**) Plots of **D1**/**MOSC-1-Co** molar ratio vs. absorbance at 360 nm based on the titration experiment. (**c**) The nonlinear form fits to the Hill equation of UV–Vis titration experiments based on the absorption band centered at 370 nm in the [**D1**]/[**MOSC-1-Co**]. Adapted with permission from [55].

**Figure 3 molecules-29-01220-f003:**
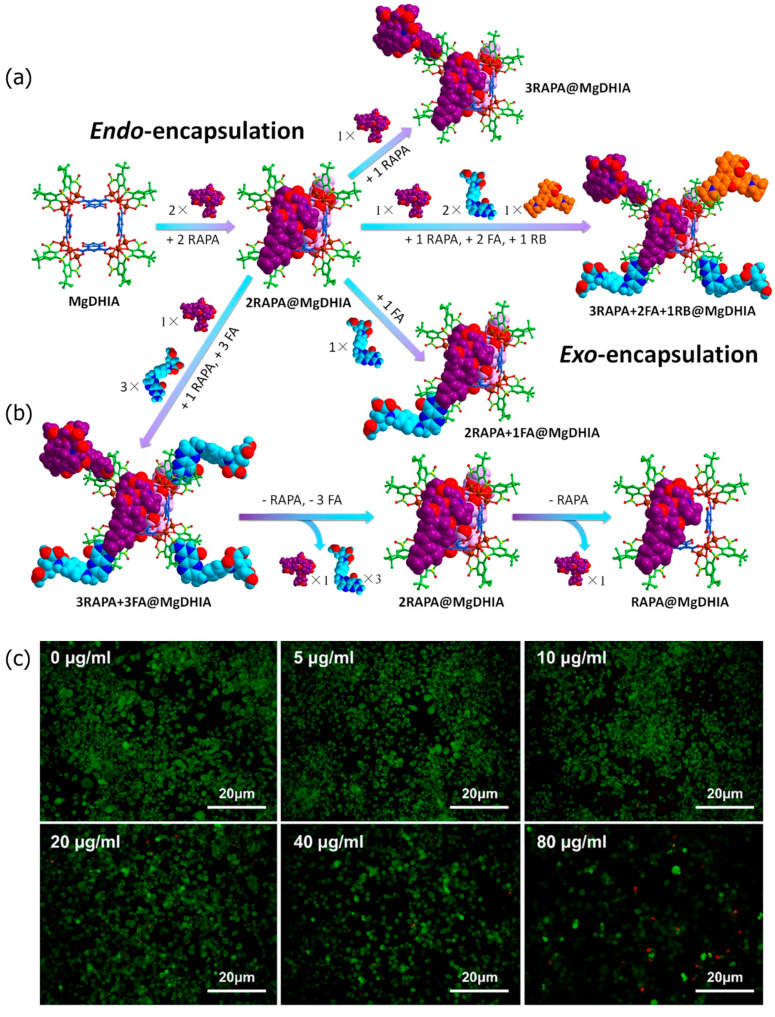
Schematic representation of (**a**) stepwise drug encapsulation by *endo* and *exo* cavities of MgDHIA; (**b**) stepwise drug-release processes of RAPA-FA@MgDHIA. (**c**) Live (green) and dead (red) staining plots of cells after 24 h of treatment with different concentrations of MgHDIA. Reproduced from [56]. Copyright (2021), with permission from Elsevier.

**Figure 4 molecules-29-01220-f004:**
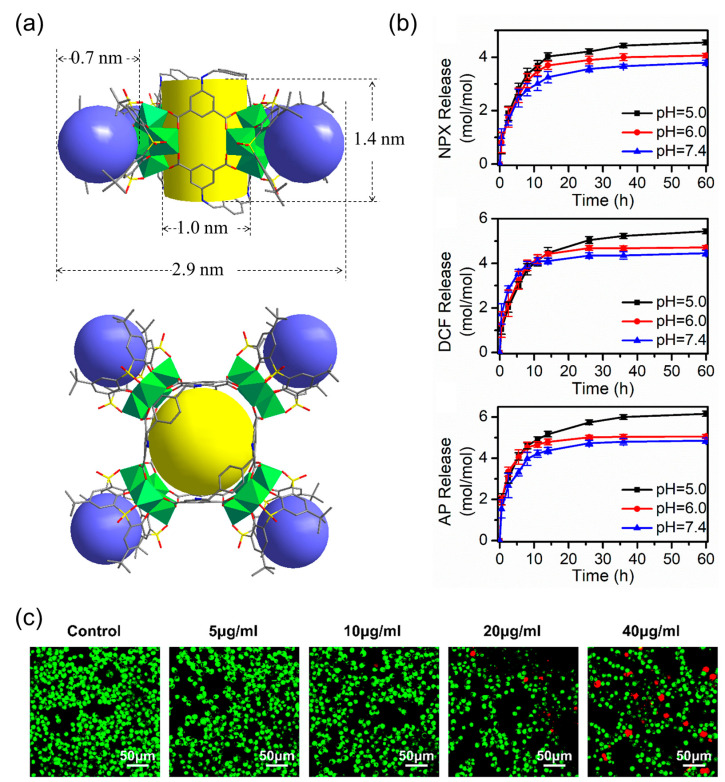
(**a**) Crystal structure of coordination container ZnPMTC. The spheres serve to guide the eyes representing the *endo* cavity (yellow) and *exo* cavities (blue). (**b**) Accumulated drug release of NPX@ZnPMTC, DCF@ZnPMTC, and AP@ZnPMTC in PBS solution at pH = 5.0, 6.0, and 7.4. (**c**) Live (green) and dead (red) staining plots of cells after 24 h of treatment with varying concentrations of ZnPMTC. Reproduced with permission from [57]. Copyright © 2021, American Chemical Society.

**Figure 5 molecules-29-01220-f005:**
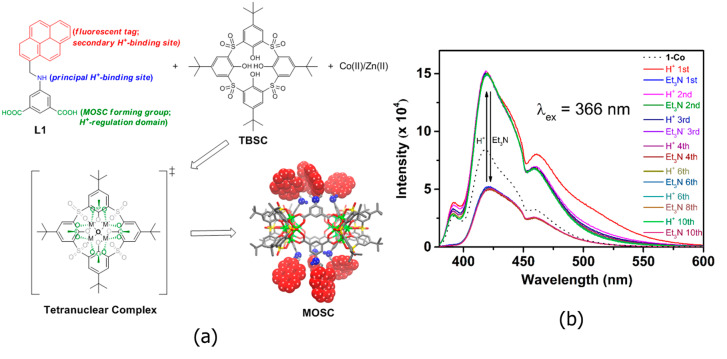
(**a**) Design of proton-responsive fluorescent MOSCs. The metal centers are shown in green, the -NH- groups are shown in blue ball, and the pyrenyl units are shown in red spacefill. (**b**) Switching “on” and “off” of Zn-NH-pyr with CF_3_COOH/Et_3_N can be repeated in multiple cycles. Reproduced with permission from [46].

**Figure 6 molecules-29-01220-f006:**
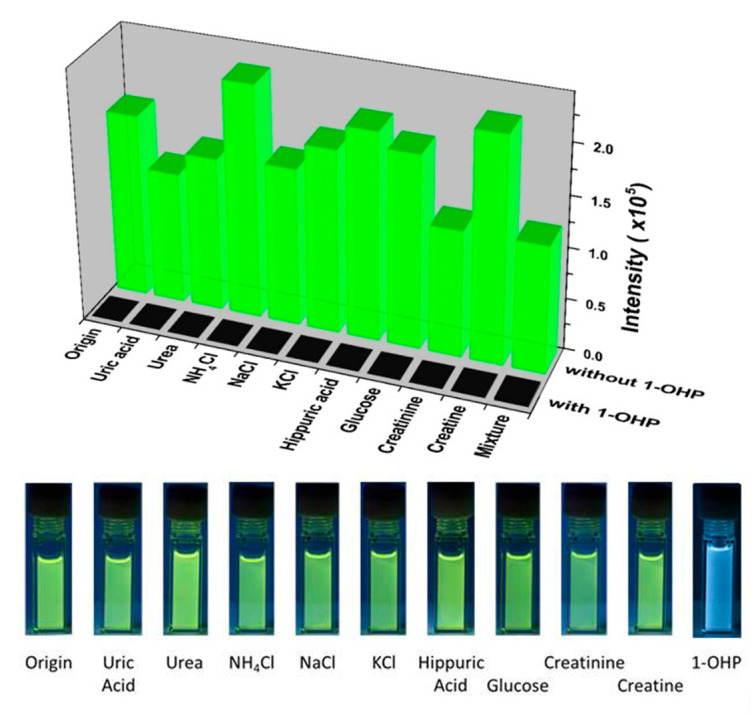
(**top**) Luminescence intensities of **Tb-TBSC** (544 nm) with various urine constituents in the CH_3_CN/H_2_O (*v*:*v* = 4:1) solution in the absence or presence of 1-OHP (*λ*_ex_ = 346 nm). (**bottom**) Photographs of the emission colors of **Tb-TBSC** with various urine constituents in the CH_3_CN/H_2_O (*v*:*v* = 4:1) solution under the irradiation of UV light at 365 nm. Reproduced with permission from [59].

**Figure 7 molecules-29-01220-f007:**
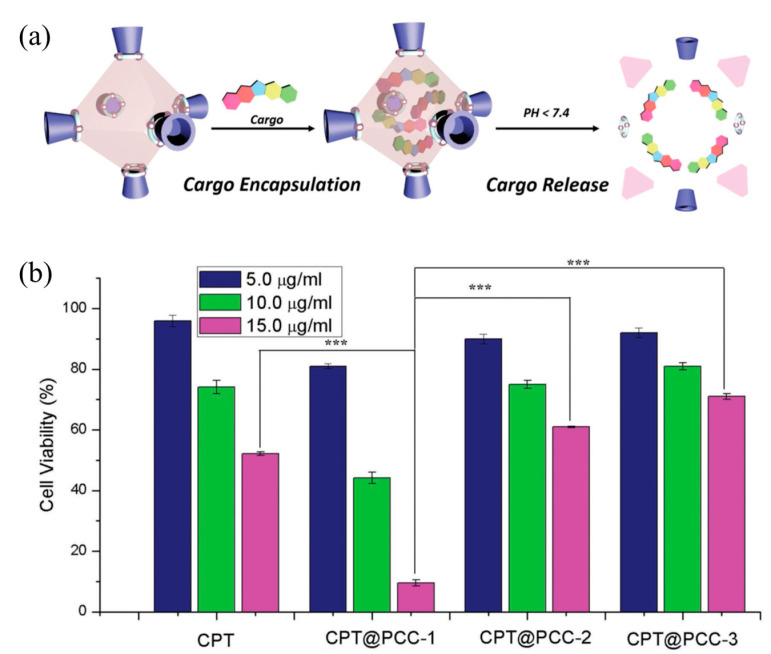
(**a**) Cargo encapsulation and release of PCC-1 and PCC-2. (**b**) Cell viability of CPT, CPT@PCC-1, CPT@PCC-2, and CPT@PCC-3 at concentration of 5.0, 10, and 15 µg mL^−1^ (*** *p* < 0.05; Student’s *t*-test), measured by Sytox Green assay. Reproduced with permission from [38]. Copyright © 2018 by John Wiley and Sons.

**Figure 8 molecules-29-01220-f008:**
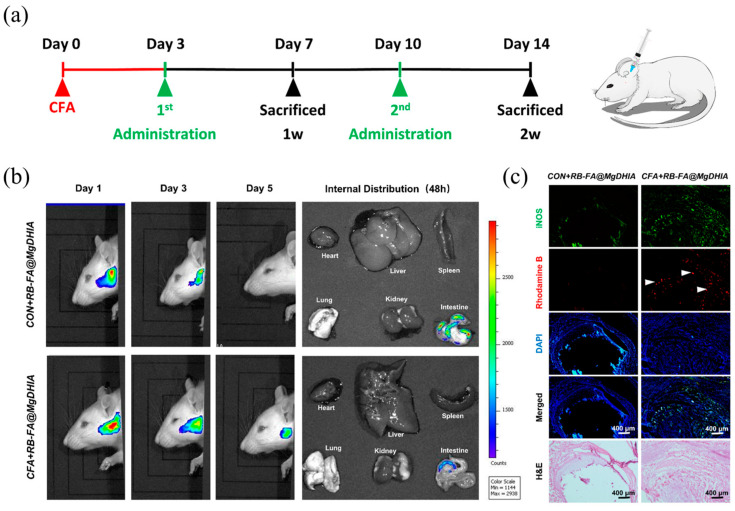
In vivo biodistribution and radiographic evaluation of Drugs@MgDHIA systems on CFA-induced TMJ inflammation. (**a**) Schematic diagram of CFA induced TMJ Inflammation in Sprague–Dawley (SD) rats. (**b**) IVIS imaging showing the biodistribution of macrophagetargeted MgDHIA (RB-FA@MgDHIA) in control and CFA groups. (**c**) Distribution of RB-FA@MgDHIA injected into the joint cavity by cryostat serial section. Reproduced with permission from [56].

**Figure 9 molecules-29-01220-f009:**
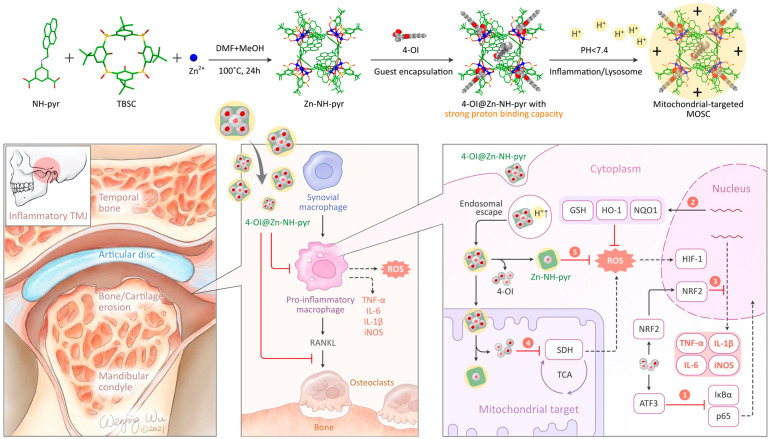
Schematic illustration of the synthesis, encapsulation, and application of 4-OI@Zn-NH-pyr for enhanced therapy in severe joint inflammation. Reproduced with permission from [58].

**Table 1 molecules-29-01220-t001:** Relevant MOSCs for biomedical applications.

MOSCs	Carboxylate Linker	Metal Ion	Sulfonylcalix[4]arene	Molecular Topology	Drug	Application	Ref
**MOSC-1-Co**	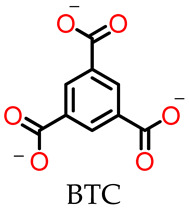	Co^2+^	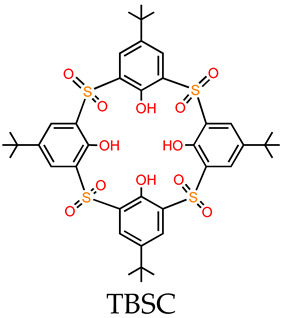	Face-directed octahedron	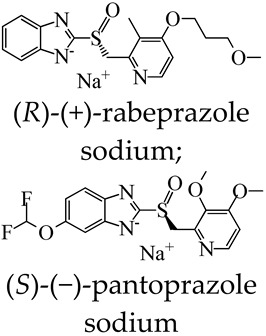	Drug encapsulation	[34]
CIAC-105	BTC	Co^2+^	TBSC	Face-directed octahedron	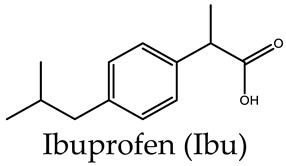	Drug loading and releasing	[37]
CIAC-106	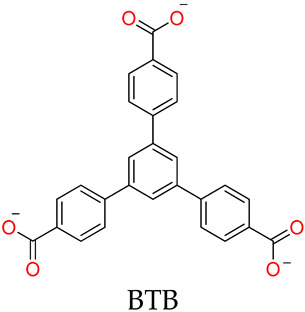	Co^2+^	TBSC	Face-directed octahedron	Ibu	Drug loading and releasing	[37]
CIAC-107	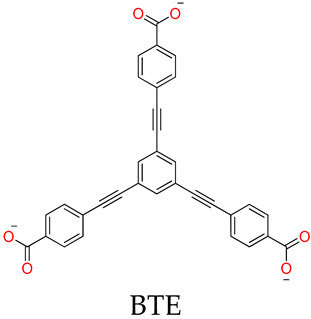	Co^2+^	TBSC	Face-directed octahedron	Ibu	Drug loading and releasing	[37]
CIAC-114	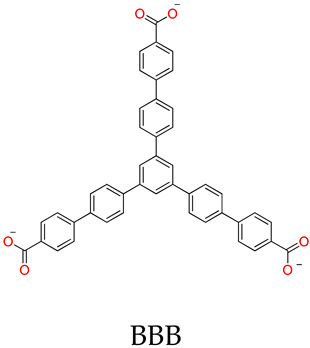	Co^2+^	TBSC	Face-directed octahedron	Ibu	Drug loading and releasing	[37]
MgDHIA	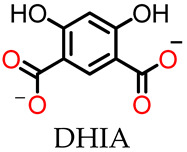	Mg^2+^	TBSC	Type-IIIbarrel-shaped box	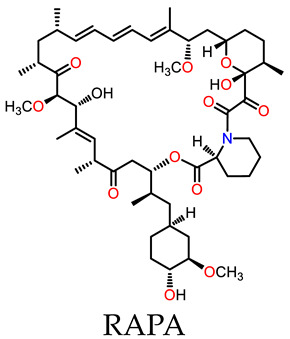 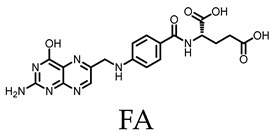	Anti-inflammatory therapy in temporomandibular joint	[56]
ZnPMTC	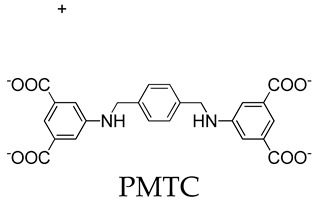	Zn^2+^	TBSC	Type-IIIbarrel-shaped box	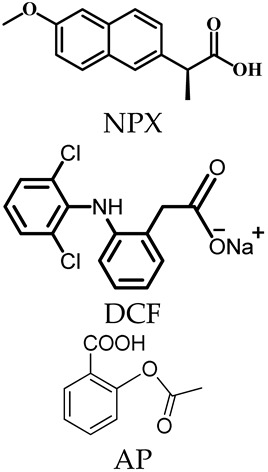	Drug loading and releasing	[57]
Co-NH-pyr	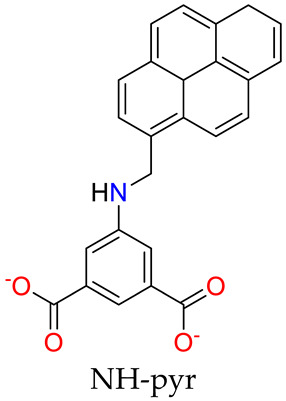	Co^2+^	TBSC	Type-IIIbarrel-shaped box		Proton sponge	[46]
Zn-NH-pyr		Zn^2+^	TBSC	Type-IIIbarrel-shaped box	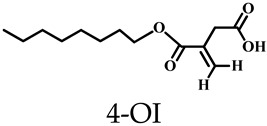	Synergistic therapy of joint inflammation	[58]
Tb-TBSC	-	Tb^3+^	TBSC	Dinuclear terbium(III) complex	-	Urinary 1-hydroxypyrene (1-OHP) detection	[59]
PCC-1	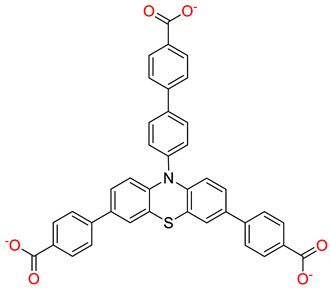	Zn^2+^	TBSC	Face-directed octahedron	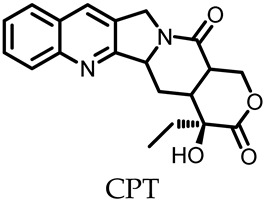	Cancer nano-therapy	[38]
PCC-2	BTB	Co^2+^	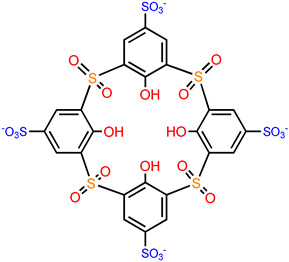	Face-directed octahedron	CPT	Cancer nano-therapy	[38]

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
