# Peer review of "Biomedical Applications of Sulfonylcalix[4]arene-Based Metal–Organic Supercontainers"

_molecules, 2024, doi:10.3390/molecules29061220_

Round 1

Reviewer 1 Report

Comments and Suggestions for Authors

The current review article extensively explores the biomedical applications of MOSCs, with a well-structured layout and clear language. I recommend publication with minor revisions to address the following points:

1. The title prominently features "Sulfonylcalix[4]arene," whereas "MOSCs" are predominantly used in the main text. I suggest revising the title for consistency.

2. Emphasizing the originality and distinguishing this review from related literature in the introduction will underscore its significance.

3. Tables could be incorporated to summarize and support the discussed studies, providing readers with a comprehensive overview of the field. This would enhance readability and understanding.

4. Adding a list of abbreviations would enhance clarity for readers.

5. On page 14, there is a sentence (line 210) containing a typo or grammatical error that requires revision.

6. Consider altering the topic of subtitle number 3.3 from "biomedical applications" to avoid repetition with the heading "3. Biomedical Applications."

7. Subtitles should be renumbered for consistency; for instance, the conclusion, currently labeled as number 5, should be numbered as 4.

Author Response

Point 1: The title prominently features "Sulfonylcalix[4]arene," whereas "MOSCs" are predominantly used in the main text. I suggest revising the title for consistency. 

Response 1: We thank the reviewer for the suggestion. The title has been revised to “Biomedical Applications of Sulfonylcalix[4]arene-Based Metal-Organic Supercontainers”.

Point 2: Emphasizing the originality and distinguishing this review from related literature in the introduction will underscore its significance.

Response 2: We thank the reviewer for this valuable comment. The sentences “Numerous reviews have provided a comprehensive overview of the the design and construction of coordination cages for biological applications [20,29-32]. At present, some critical reviews on the synthesis and application of MOSCs have been published [24,25,33], however, there is no review exclusively focusing on the bio-application of MOSCs. Thus, this article gives a brief introduction to the MOSCs as well as their host-guest chemistry, and highlights significant recent developments in biomedical applications of MOSCs. We aim at providing a clear and comprehensive understanding for this exciting research field to promote further development and applications.” have been added in the section of Introduction.

Point 3: Tables could be incorporated to summarize and support the discussed studies, providing readers with a comprehensive overview of the field. This would enhance readability and understanding.

Response 3: Table 1 has been added in the manuscript to summarize the relevant MOSCs for biomedical applications.

Point 4: Adding a list of abbreviations would enhance clarity for readers.

Response 4: A list of abbreviations has been added in the section of “Abbreviations” at the end of manuscript.

Point 5: On page 14, there is a sentence (line 210) containing a typo or grammatical error that requires revision.

Response 5: The sentence has been revised to “Dai and co-workers [51] reported the design and assembly of a new MOSC (ZnPMTC) …”

Point 6: Consider altering the topic of subtitle number 3.3 from "biomedical applications" to avoid repetition with the heading "3. Biomedical Applications."

Response 6: The subtitle of section 3.3 has been revised to “3.3 Targeted drug delivery and therapeutic effect”.

Point 7: Subtitles should be renumbered for consistency; for instance, the conclusion, currently labeled as number 5, should be numbered as 4.

Response 7: The subtitle of conclusion section has been revised to “4. Conclusion and outlook”.

Reviewer 2 Report

Comments and Suggestions for Authors

The manuscript by Chen, Dai and coworkers describes the progress in the field of biomedical applications of sulfonylcalix[4]arene coordination cages. The review is well-organized, and perhaps in the introduction, a bit more context is needed by citing other significant works and reviews in the field to contextualize the family of sulfonylcalix[4]arene coordination cages within the broader field of molecular cages.

The authors have effectively organized the review by first describing the different compounds, indicating the main families, their synthesis, and properties. Additionally, they have provided a detailed analysis of the cavity of each cage family, indicating the volume of the cavity. The authors need to clarify some nomenclature. For instance, it's not clear in this phrase "two wide openings (ca. 7Å × 7 Å), a wider inner dimension (ca. 10 Å × 10 Å), and a deep pocket (with a depth of ca. 12 Å)" what each of these dimensions means. This should be clearly indicated in Figure 1. Figure 1 should also indicate which metals are used to prepare this type of molecular cages. Additionally, it should outline typical reaction conditions, including reaction time, temperature, solvent, and yield of these reactions. This is crucial because metals inherently possess toxicity, especially in biological applications. Therefore, the authors should describe which metals are used, emphasizing their low toxicity for biological applications. This information should be described both in the text and in Figure 1.

Regarding biomedical applications, the authors first describe the encapsulation of drugs within the molecular cages, conducting an analysis of both the interior and exterior cavities. Section 3.1 Drug Encapsulation, should be renamed, as it also includes compounds that are not drugs, such as polycyclic aromatic compounds framed as carcinogenic elements and therefore not drugs. The examples provided of biomedical applications are well-described with representative images. In this regard, the authors need to review the phrase "reproduced with permission from the reference" as they should also indicate the original copyright of the figure. The authors must thoroughly review the exact phrase required to reproduce a figure. In Figures 6, 7, 8, the authors need to include the chemical structures of the molecular cages and the drugs used in the studies. It is necessary to include the structures to quickly evaluate which type of molecular cage and which drugs have been used in each study. This change is crucial for providing context and facilitating understanding for the readers. It's also important for the authors to indicate the type of molecular cage in the examples described, relating it to the nomenclature described in Figure 1. Another aspect that the authors should improve is indicating the toxicity of the molecular cages against cells, i.e., molecular cages for use in biological applications should have low or very low toxicity. To address this, the authors should explicitly state the toxicity of each example included in the review.

The references need to be unified since in some cases the DOI appear as links and in others they don't. I believe it's most useful for readers if DOI appear as links so they can directly access the referenced articles.

Overall, the review is well-written, with representative examples that are highly valuable for individuals working in the field of molecular cages. Therefore, I recommend the publication of this article after addressing the issues I have highlighted, including the permission sentence for the figures and conducting a detailed review of the English language throughout the article.

Comments on the Quality of English Language

Overall, the article is well-written, but some sentences are difficult to understand. Therefore, it is recommended that the authors carefully review the writing of the article to rewrite some long sentences and improve readability. Additionally, they should thoroughly review the grammar and spelling of the words.

Author Response

Point 1: The review is well-organized, and perhaps in the introduction, a bit more context is needed by citing other significant works and reviews in the field to contextualize the family of sulfonylcalix[4]arene coordination cages within the broader field of molecular cages. 

Response 1: We thank the reviewer for this valuable comment. The sentences “Numerous reviews have provided a comprehensive overview of the the design and construction of coordination cages for biological applications [20,29-32]. At present, some critical reviews on the synthesis and application of MOSCs have been published [24,25,33], however, there is no review exclusively focusing on the bio-application of MOSCs. Thus, this article gives a brief introduction to the MOSCs as well as their host-guest chemistry, and highlights significant recent developments in biomedical applications of MOSCs. We aim at providing a clear and comprehensive understanding for this exciting research field to promote further development and applications.” have been added in the section of Introduction.

Point 2: The authors have effectively organized the review by first describing the different compounds, indicating the main families, their synthesis, and properties. Additionally, they have provided a detailed analysis of the cavity of each cage family, indicating the volume of the cavity. The authors need to clarify some nomenclature. For instance, it's not clear in this phrase "two wide openings (ca. 7Å × 7 Å), a wider inner dimension (ca. 10 Å × 10 Å), and a deep pocket (with a depth of ca. 12 Å)" what each of these dimensions means. This should be clearly indicated in Figure 1. Figure 1 should also indicate which metals are used to prepare this type of molecular cages. Additionally, it should outline typical reaction conditions, including reaction time, temperature, solvent, and yield of these reactions. This is crucial because metals inherently possess toxicity, especially in biological applications. Therefore, the authors should describe which metals are used, emphasizing their low toxicity for biological applications. This information should be described both in the text and in Figure 1.

Response 2: We thank the reviewer for this valuable comment. The Figure 1 has been updated with the information of chemical structure of sulfonyl-calix[4]arene, reaction conditions, and type of metal ions. The sentence “The 1,3-benzenedicarboxylate (1,3-BDC) directed Type-III MOSCs are composed of four tetranuclear units double-bridged by eight 1,3-BDC linkers, and possess four exo cavities and a barrel shaped endo cavity featuring two wide openings (ca. 7Å × 7 Å), a wider inner dimension (ca. 10 Å × 10 Å), and a deep pocket (with a depth of ca. 12 Å) [39].” has been revised to “The 1,3-benzenedicarboxylate (1,3-BDC) directed Type-III MOSCs are composed of four tetranuclear units double-bridged by eight 1,3-BDC linkers, and possess four exo cavities and a barrel shaped endo cavity featuring two wide openings (ca. 7Å × 7 Å defined by the shortest distances between two opposite carboxylate linkers), a wider inner dimension (ca. 10 Å × 10 Å defined by the longest distances between two opposite carboxylate linkers), and a deep pocket (with a depth of ca. 12 Å) [39].

Point 3: Regarding biomedical applications, the authors first describe the encapsulation of drugs within the molecular cages, conducting an analysis of both the interior and exterior cavities. Section 3.1 Drug Encapsulation, should be renamed, as it also includes compounds that are not drugs, such as polycyclic aromatic compounds framed as carcinogenic elements and therefore not drugs.

Response 3: We thank the reviewer for the valuable suggestion. The subtitle of section 3.1 has been revised to “3.1. Host-guest chemistry and drug encapsulation”.

Point 4: The examples provided of biomedical applications are well-described with representative images. In this regard, the authors need to review the phrase "reproduced with permission from the reference" as they should also indicate the original copyright of the figure. The authors must thoroughly review the exact phrase required to reproduce a figure.

Response 4: The statements of reproducing images from references have been corrected.

Point 5: In Figures 6, 7, 8, the authors need to include the chemical structures of the molecular cages and the drugs used in the studies. It is necessary to include the structures to quickly evaluate which type of molecular cage and which drugs have been used in each study. This change is crucial for providing context and facilitating understanding for the readers. It's also important for the authors to indicate the type of molecular cage in the examples described, relating it to the nomenclature described in Figure 1.

Response 5: Table 1 including the structural information of relevant MOSCs for biomedical applications has been added in the manuscript.

Point 6: Another aspect that the authors should improve is indicating the toxicity of the molecular cages against cells, i.e., molecular cages for use in biological applications should have low or very low toxicity. To address this, the authors should explicitly state the toxicity of each example included in the review.

Response 6: The information of safe dose, biocompatibility, and toxicity of MOSCs used in biomedical applications has been added in manuscript.

Point 7: The references need to be unified since in some cases the DOI appear as links and in others they don't. I believe it's most useful for readers if DOI appear as links so they can directly access the referenced articles.

Response 7: The references have been revised with hyperlinks added for DOI.
